

# Past and future response of the North Atlantic warming hole to anthropogenic forcings

Saïd Qasmi

CNRM, Université de Toulouse, Météo-France, CNRS, Toulouse, France

**Correspondence:** Saïd Qasmi (said.qasmi@meteo.fr)

**Abstract.**

Most of the North Atlantic ocean has warmed over the last decades, except a region located over the subpolar gyre, known as the North Atlantic warming hole, where sea surface temperature has in contrast decreased. Previous assessments have attributed part of this cooling to the anthropogenic forcings (aerosols and greenhouse gases) modulated by decadal internal variability. Here I use an innovative and proven statistical method which combines climate models and observations to confirm the anthropogenic role in the WH cooling, and to provide estimates of the contribution of a set of given external forcings. Furthermore, the method is able to reduce the uncertainty in the WH temperature over the historical period, but also in the future, with a decrease of 65% in the short term, up to 50% in the long term. A model evaluation validates the reliability of the obtained projections. In particular, the projections associated with a strong temperature increase over the warming hole are now excluded from the likely range obtained after applying the method.

## 1 Introduction

The increase of global surface air temperature over the 1850-2021 period is unequivocally attributed to human activities (Lee et al., 2021). This increase is spatially heterogeneous, with some regions warming less rapidly than others, or even cooling. Noticeably, sea surface temperature (SST) over the so-called North Atlantic "warming hole" (hereafter WH) region, which is located over the subpolar gyre, has decreased over the 1901-2021 period relative to the 1870-1900 period (Drijfhout et al., 2012). This long-term decrease is often associated with modifications in the meridional transport of oceanic heat content in the North Atlantic (Gervais et al., 2018; Hu and Fedorov, 2020), melting of the Arctic sea ice and the Greenland ice cap (Allan and Allan, 2019; Gervais et al., 2018; Keil et al., 2020; Liu et al., 2019), both related to a slowing down of the Atlantic Meridional Overturning Circulation (AMOC) (Rahmstorf et al., 2015; Caesar et al., 2018). Atmospheric processes have also been proposed to explain this cooling (Li et al., 2022; Keil et al., 2020). Although internal variability strongly influences interannual to decadal variability in the North Atlantic SST (Robson et al., 2016; Hodson et al., 2014; Moffa-Sánchez et al., 2019), several detection and attribution studies indicate a contribution of anthropogenic greenhouse gases (GHG) to explain the long-term cooling of the WH over the historical period (Chemke et al., 2020; Dagan et al., 2020). Concomitantly, natural and especially anthropogenic aerosols may have played a role in shaping the temporal evolution of the WH SST (Booth et al., 2012; Fiedler and Putrasahan, 2021), by delaying the WH cooling through an acceleration of the AMOC associated with a warming of the WH (Dagan et al.,



2020; Menary et al., 2020). Climate projections from the fifth Coupled Model Intercomparison Project (CMIP5) and from CMIP6 (Eyring et al., 2016) corroborate the contribution of GHGs, with some models projecting an enhanced cooling over the WH (Dagan et al., 2020; Sigmond et al., 2020; Marshall et al., 2015), but with large uncertainties in the spatio-temporal structure and intensity of the cooling (Menary and Wood, 2018; Bellomo et al., 2021). Given the potential impact of the North

Atlantic SSTs on the North hemisphere climate (Qasmi et al., 2021; Ren and Liu, 2021; Gervais et al., 2019, 2020; Karnauskas et al., 2021), it is important to investigate to what extent it is possible to reduce the uncertainty in the SST projections over the WH.

So far, the impact of external forcings and internal variability on the observed WH over the historical period has been qualitatively estimated by using dedicated sensitivity model experiments (Sigmond et al., 2020) or initial-condition large ensembles

(Dagan et al., 2020). A recent detection and attribution study based on a statistical method derived from optimal fingerprinting provides a quantitative statement regarding an anthropogenic signal in the WH SST time series (Chemke et al., 2020), but without quantifying the influence of each external forcing. In this paper, I estimate (i) the contribution of each of the main external forcings in the evolution of the observed WH over the historical period, and (ii) the uncertainty on the evolution of the WH response to external forcings (hereafter forced response) both over the historical and future periods. These estimates come

from the Kriging for Climate Change method, an innovative and proven statistical method, which combines observations and climate models.

## 2  Data and Methods

### 2.1  Observations

SST observations from the HadSST4 data set (Kennedy et al., 2019) are used. These observations have the advantage of

providing an estimate of the measurement uncertainty, which encompasses the treatment of incomplete data coverage, homogenization uncertainty etc. through an ensemble of 200 equiprobable realisations over the 1870-2021 period. Compared to other datasets, this ensemble allows a more comprehensive estimation of the measurement error with the possibility of calculating the temporal covariance of uncertainty with the 200 members. Note that I do not consider other datasets because they usually fall within the uncertainty of HadSST4. The median of the 200-member ensemble will be considered as the best-estimate,

while all realisations are used to estimate the measurement uncertainty. Observations of global surface air temperature (GSAT) used by the KCC method are taken from the HadCRUT5 ensemble of 200 members (Morice et al., 2021).

### 2.2  Climate Models

Historical and ScenarioMIP simulations from the CMIP6 ensemble (Eyring et al., 2016; O'Neill et al., 2016) are used to estimate the forced response of temperature to all external forcings over the 1850-2100 period (the 3 scenarios SSP1-2.6,

SSP2-4.5 and SSP5-8.5 are used). The contribution of each external forcing during the 1850-2020 period is estimated from the DAMIP ensemble (Gillett et al., 2016), especially from the hist-GHG and hist-AER simulations, in which GHGs and





anthropogenic aerosols follow their historical concentrations, respectively, while other forcings are kept constant. The list of the models and the simulations used is provided in Table A1. A particular feature of some of the CMIP6 models, on which the sixth assessment report (AR6) from the IPCC is based, is their high equilibrium climate sensitivities (ECS) (Lee et al., 2021).

In the AR6, ECS and GSAT projections from the CMIP6 ensemble have been assessed by using statistical methods (including the one used in this study) and observations to provide GSAT estimates consistent with the observational record (Ribes et al., 2021). The statistical method used here takes these likely estimates of ECS and GSAT future ranges results into account, by including an observational constraint based on both GSAT and SST over the WH region.

## 2.3  Statististical method

To assess past and future forced response of the WH, I use an observational constraint method that has been previously applied to global mean warming (Ribes et al., 2021) and regional warming (Qasmi and Ribes, 2021; Ribes et al., 2022). This technique called Kriging for Climate Change (KCC) works in 3 steps. First, the forced response of each climate model is estimated over the historical period. In order to also get attribution statements, the responses to ALL (all external forcings), NAT (natural forcings only) and anthropogenic GHG forcings are estimated separately. Second, the sample of the forced responses from

the CMIP6 models is used as a prior of the real-world forced response. This is done assuming that "models are statistically indistinguishable from the truth". Third, observations are used to derive a posterior distribution of the past and future forced response given observations, in a Bayesian way. The procedure can be summarized using the following equations:

$$y = Hx + \epsilon, \tag{1}$$

where $y$ is the time-series of observations (a vector), $x$ is the time-series of the forced response (a vector), $H$ is an obser-

vational operator (matrix), $\epsilon$ is the random noise associated with internal variability and measurement errors (a vector), and $\epsilon \sim \mathcal{N}(0, \Sigma_y)$, where $\mathcal{N}$ stands for the multivariate Gaussian distribution. The observational operator $H$ is a matrix which extracts the components of $x$ that are observed in $y$, thus all $H$ coefficients are equal to 0 or 1 (see Eq. (B1)). Climate models are used to construct a prior on $x$: $\Pi(x) = \mathcal{N}(\mu_x, \Sigma_x)$. Then the posterior distribution given observations $y$ can be derived as $p(x|y) = \mathcal{N}(\mu_p, \Sigma_p)$. Remarkably, $\mu_p$ and $\Sigma_p$ are available in closed-form expressions. In the following, I am interested in

assessing the forced response of annual mean SST over the WH (projections), as well as the annual mean response to specific subsets of forcings (attribution). These forced responses could be constrained by various observations. Here, I consider constraints by both GSAT observations and regional annual mean SST (averaged over the WH) – the rationale behind this choice is discussed below. Therefore,

$$x = (T^{\mathrm{all}}_{\mathrm{glo}}, T^{\mathrm{all}}_{\mathrm{reg}}, T^{\mathrm{ghg}}_{\mathrm{reg}}, T^{\mathrm{nat}}_{\mathrm{reg}}), \tag{2}$$

where each element is the annual forced response over the 1850–2100 period (except $T^{ghg}_{reg}$ which only covers 1850–2020). $T$ stands for temperature, "all", "ghg" or "nat" are the subsets of external forcings considered, "glo" and "reg" refer to GSAT and regional SST, respectively. The length of $x$ is thus $n_x = 924$. Similarly,





$$\boldsymbol{y} = (\boldsymbol{T}_{\mathrm{glo}}^{\mathrm{obs}}, \boldsymbol{T}_{\mathrm{reg}}^{\mathrm{obs}}), \tag{3}$$

i.e., only observed time-series are used in $\boldsymbol{y}$. The length of these time-series varies: 1850-2021 for GSAT, 1870-2021 for
the WH index. As a result, $\boldsymbol{y}$ has a length of $n_y = 322$. All attribution or projection diagnoses presented below can be derived
from the posterior distribution $p(\boldsymbol{x}|\boldsymbol{y})$. Accounting for GSAT is important because various recent studies argued that the
observational constraint on this variable is robust (eg, to the choice of the statistical method), with the high-end of simulated
GSAT model responses not consistent with observed GSAT changes (Lee et al. (2021), and references therein). As there is a
some dependence (between CMIP models) between future GSAT changes and regional changes over most regions including
the WH, a reduced GSAT response is expected to imply a reduced regional warming. This is confirmed by Qasmi and Ribes
(2021), who found that accounting for the global constraint clearly improves the accuracy of regional projections. Accounting
for regional observations is also relevant, especially over regions where long observational records are available and the climate
change signal has already emerged. Qasmi and Ribes (2021) also report a significant added-value in doing so. The data to be
included in the observational constraint represent a key element of the proposed method which is further discussed.

Implementing this methodology requires to compute the values of $\boldsymbol{\mu}_{\mathrm{x}}$, $\boldsymbol{\Sigma}_{\mathrm{x}}$, and $\boldsymbol{\Sigma}_{\mathrm{y}}$. Following Qasmi and Ribes (2021), $\boldsymbol{\mu}_{\mathrm{x}}$
and $\boldsymbol{\Sigma}_{\mathrm{x}}$ are estimated as the sample mean and covariance of the CMIP6 model forced responses. $\boldsymbol{\Sigma}_{\mathrm{y}}$ requires statistical mod-
elling of internal variability and measurement uncertainty. HadSST4 and HadCRUT5 ensembles are to estimate measurement
uncertainty. For internal variability within the GSAT and WH time series, I follow Qasmi and Ribes (2021) in using a mixture
of Auto-Regressive processes of order 1 (MAR). This model allows to capture interannual to decadal internal variations within
the observed time series. In order to assess internal variability, a usual technique is to consider the residuals of the difference
between the CMIP6 multimodel mean (the forced response estimate) and the observation time series. However, these residuals
are likely biased at the regional scale as the forced response estimated by the multimodel mean is not necessarily consistent
with the observations. Instead, in order to obtain a robust estimate of the internal variability, an iterative algorithm is applied
so that internal variability before and after the constraint remains consistent with the forced response (see Eq. D3). In addition,
the dependance between global and regional internal variability is taken into account by accounting for the covariance between
the regional vs global residuals in the MAR modelling. Further details and discussion about the structure of $\boldsymbol{H}$, the estimation
of the inputs parameters $\boldsymbol{\Sigma}_{\mathrm{x}}$ and $\boldsymbol{\Sigma}_{\mathrm{y}}$, are provided in the Appendices. Note that I do not account for uncertainty in the input
parameters $\boldsymbol{\mu}_{\mathrm{x}}$, $\boldsymbol{\Sigma}_{\mathrm{x}}$, and $\boldsymbol{\Sigma}_{\mathrm{y}}$. – this could be done by using more complex hierarchical Bayesian models.

## 3   Results and discussion

### 3.1   Attribution of the observed warming hole

Historical simulations and single-forcing experiments from the DAMIP ensemble give a first characterization of the WH and
an estimate of the contribution of each external forcing. Over the 1951-2014 period, anthropogenic aerosols have contributed
to warm the subpolar gyre, with a mean increase of +0.5°C (Fig. 1a) compared to the 1870-1900 period. Concomitantly, the





effect of the GHGs is a cooling over the same region, with a mean decrease of about -0.2°C (Fig. 1b). Note that similar anomaly

patterns are obtained when using the pre-industrial control simulations as a reference (Fig. S1). These anomalies support the

conclusions of several studies and may reflect the signature of a modification of the oceanic heat transport (Dagan et al., 2020),

rather than a direct radiative impact from the aerosols (via a parasol effect) or from the GHGs (warming the surface). The

impact of the natural forcings on the WH SST is weak in the models, with a slight warming over the Labrador Sea (Fig. 1c).

Historical simulations from models which have contributed to the DAMIP ensemble show a partial compensation of the effects

induced by anthropogenic aerosols and GHGs, resulting in a warming over the subpolar gyre and a slight cooling over the Gulf

Stream region (Fig. 1d). A similar pattern is found when historical simulations from all CMIP6 models are considered (Fig. S2).

This result from the historical simulations is not consistent with the observations, which indicate a -0.4°C cooling over a large

area of the subpolar gyre (Fig. 1e) and resembles more closely the GHG multi-model mean. This difference between historical

simulations and observations could be explained by non-exclusive factors : (i) biases in the simulation of the physical processes

driving the SST variability over the subpolar gyre, (ii) an overestimated impact of aerosols in the models, as pointed out by

some studies, (iii) an underestimated impact of GHGs, (iv) internal variability. In any case, the CMIP6 multi-model mean does

not reflect the diversity of the SST variability in the North Atlantic since the location and the intensity of the anomalies over

the historical period varies considerably in this ensemble (Fig. S3), a feature also existing in previous generations of models

(Ba et al., 2014; Menary and Wood, 2018).

## Average over 1951-2014 wrt 1870-1900

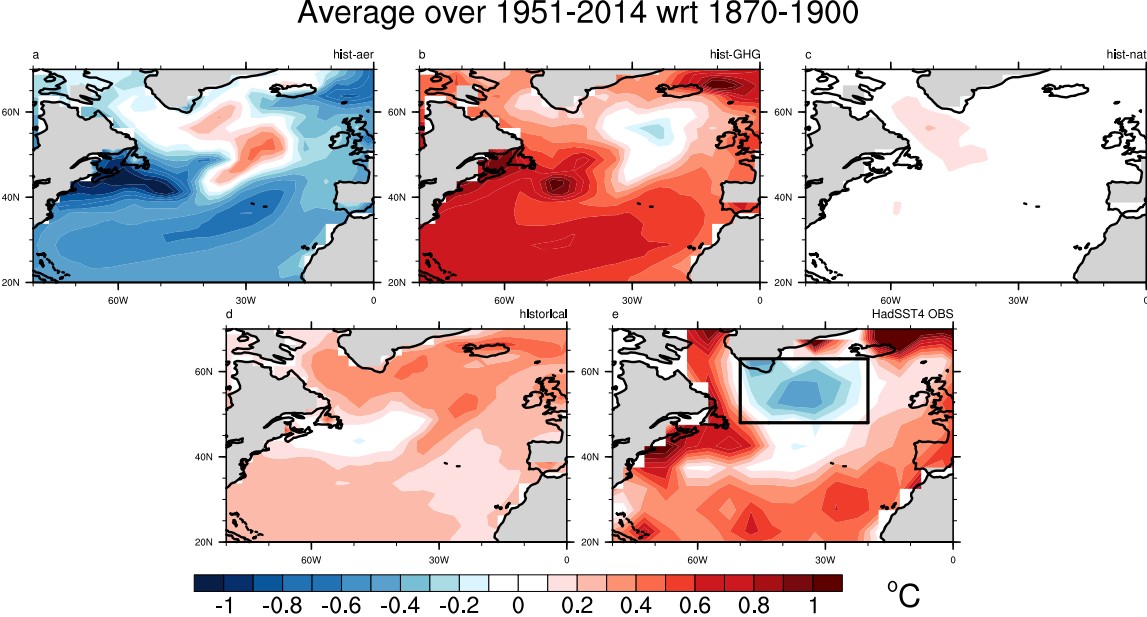

**Figure 1.** (a) Annual SST multi-model mean difference between the 1951-2014 and 1870-1900 periods for the CMIP6 hist-aer simulations.
(b) Same as (a) but for hist-GHG. (c) Same as (a) but hist-nat. (d) Same as (a) but for historical. (e) HadSST4 SST annual anomalies over the
1951-2014 period relative to the 1870-1900 period.



The combination of CMIP6 models and observations via the KCC statistical method provides estimates of the SST forced
responses to different external forcings with uncertainties that are consistent with the available observations and internal vari-
ability. Here, I apply the KCC method to the DAMIP ensemble by using a WH index, defined as the spatial average over
48°N;63°N and -50°W;-20°W, of the annual SST over the 1870-2021 period (see black domain in Fig. 1e). As a first step, this
domain is common to all models.

Figure 2 shows the attribution results over the 1951-2020 period compared to the 1870-1900 period. The contribution of all
external (ALL) forcings is a cooling of the WH, with a best estimate of $-0.14 \pm 0.12°$ C, which is half of the observed cooling
$(-0.3 \pm 0.12°$ C). The constrained ALL response is opposite in sign to the unconstrained response from the CMIP6 ensemble,
which indicates an underestimation of the cooling, consistent with Fig. 1de. The uncertainty in the ALL response is reduced by
about 70% compared to the unconstrained response. The contributions from natural (NAT) and anthropogenic (ANT) forcings

(contributing to ALL) are examined. The response due to natural NAT forcings has a very small contribution, with a slight
warming of $+0.06 \pm 0.02°$ C. Note that this estimate is consistent with the hist-nat simulations. The major contribution of
the ALL-induced cooling comes from the ANT component, with a cooling of $-0.24 \pm 0.12°$ C and a reduction of 70% in the
uncertainty, which now excludes positive values. This decrease in uncertainty is almost equally distributed between the GHG
response and the response to other anthropogenic (OA, including aerosols, ozone...) forcings, with a decrease in uncertainty

of about 65% in both components. Note that the unconstrained OA response (estimated as the ANT minus GHG difference) is
consistent with the hist-aer ensemble (Fig. S4). As the uncertainty in the constrained GHG and OA terms is still sampling both
positive and negative values, it is not possible to provide a clear contribution of these forcings over the historical period.





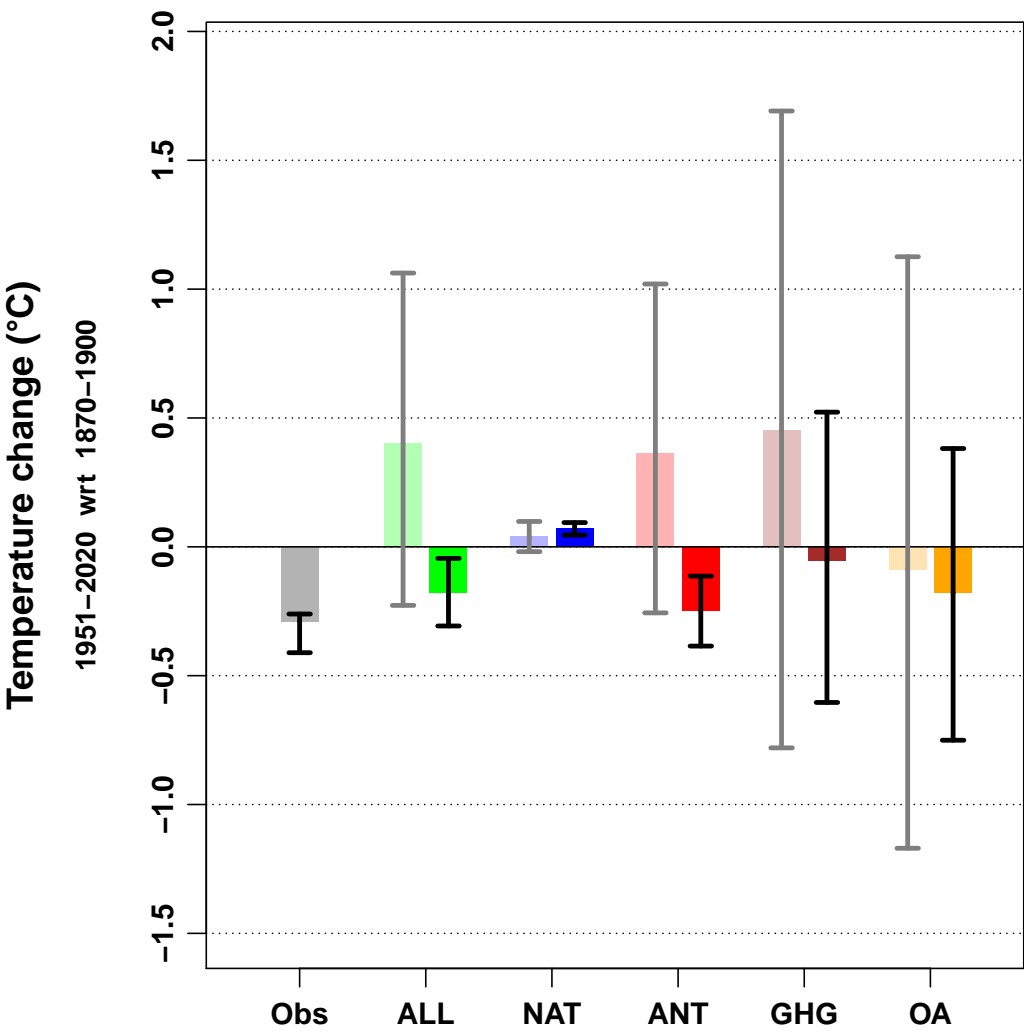

**Figure 2.** Changes induced by various subsets of external forcings over the historical period [1951–2020 with respect to (wrt) 1870–1900]. Observed (Obs) changes (gray, left) are deduced from HadSST4 observations; uncertainty only includes observational uncertainty (i.e., measurement and processing; internal variability is ignored). For all other contributions, the left handside bar and gray confidence interval describe the DAMIP model range, assuming a Gaussian distribution. The right handside bar and black confidence interval correspond to results constrained by observations. All ranges shown are 5 to 95% confidence ranges. The SSP2-4.5 scenario is used to extend historical simulations after 2014.

The sensitivity of these attribution results to methodological factors is quantified by considering several aspects. First, to take into account the diversity of the CMIP6 models in the spatial structure of the simulated WH, a WH index is defined for each model over a specific domain, to ensure a consistent comparison with observations. As done by Menary and Wood (2018), each WH model index is based on a domain where a noticeable warming over the subpolar gyre in the hist-aer simulations





is colocated with a non-warming (or cooling) area in the hist-GHG simulation over the 1951-2021 period (see black boxes in figures S5-S6). Note that this ad hoc selection is relevant since for all models, the anomalies are co-located with the climatology estimated from the piControl simulations for most models (Fig. S7). Figure S8 shows that considering this adaptive definition of the WH index changes the best-estimate values of the GHG and OA response, but with an uncertainty spread almost as large as in Fig. 2, meaning that the conclusions stated above do not change. The second sensitivity test is to consider the full set of CMIP6 models available to estimate the forced responses, such that a sample of 27 models is considered instead of the only 12 models that contributed to the DAMIP ensemble. Using the KCC method, hist-GHG-like simulations for the models that did not contribute to the DAMIP ensemble are reconstructed through the 1%-$CO_2$ simulations (in which the $CO_2$ concentration increases by 1% each year for 150 years), as done by Ribes et al. (2021) (see their Supplementary Material 1.4). The inclusion of all CMIP6 models slightly the uncertainty ranges, but does not change the main conclusions about the attribution of the observed WH cooling (Fig. S9).

## 3.2 Constraining projections

The KCC method is also used to constrain the SST projections associated with the CMIP6 SSPs simulations over the WH. Since DAMIP simulations are not required to perform the calculations based only on the ALL component, all available CMIP6 models are used to estimate the past and future forced responses, using a fixed WH domain as in Fig. 2. Figure 3 shows that the uncertainty in the three selected scenarios is reduced by 65% for a mid-time period (2041-2060), and by 50% at the end of the 21st century compared to the unconstrained projections. The lower bound of the projections distributions is less affected than the upper bound. For example, in 2100 for the SSP2-4.5 simulation, the constrained lower bound of -0.76°C is slightly revised upwards compared to the unconstrained bound (-2.08°C), while the upper bound is revised downwards considerably (+2.34°C and +4.59°C for the constrained and unconstrained values, respectively). Consequently, trajectories associated with a very strong warming of the WH are excluded from the posterior distribution (i.e. after the constraint). Note that this posterior distribution over the projected period largely surrounds the 0°C value. This indicates that even if a considerable warming of the WH is less likely in the future, projections remain uncertain about the sign relative to the future SST changes.


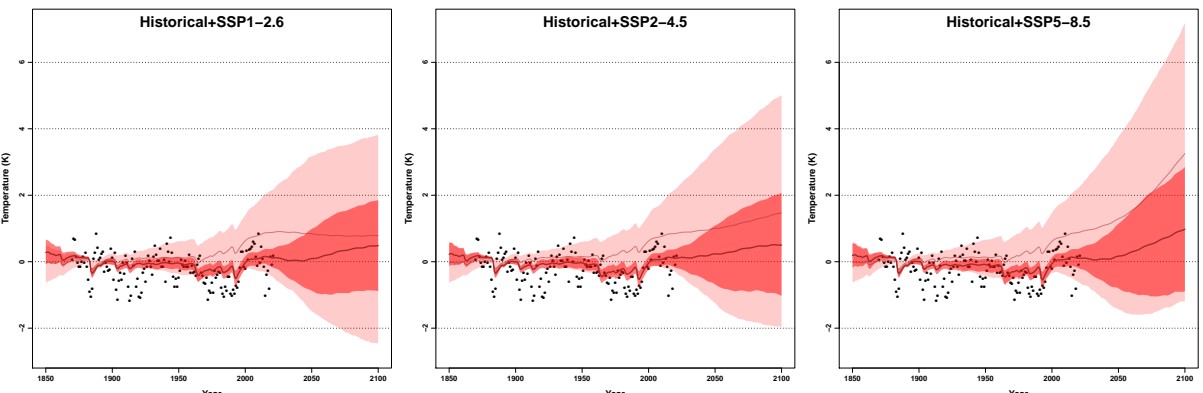

**Figure 3.** The observational constraint is applied to concatenated historical and SSP scenarios simulations (SSP1-2.6, SSP2-4.5, or SSP5-8.5). Annually observed values of WH SST (black points) are compared to the unconstrained (pink) and constrained (red) 5 to 95% confidence ranges of forced response as estimated from 27 CMIP6 models. All temperatures are anomalies with respect to the period 1870–1900.

In order to evaluate the confidence these results, the KCC method is applied within a perfect model framework, following a leave-one-out cross-validation:

- for a given model, a single member is considered as pseudo-observations $\boldsymbol{y}$ over the 1850–2021 period (the historical simulation is extended by the SSP2-4.5 simulation over the 2015-2021 period).

- the other 26 models are used to derive the prior distribution $\Pi(\boldsymbol{x}) \sim \mathcal{N}(\boldsymbol{\mu}_{\mathrm{x}}, \boldsymbol{\Sigma}_{\mathrm{x}})$.

- as done with the real observations, internal variability within the pseudo-observations is estimated from the difference between the pseudo-observations time series and the forced temperature response estimated by the ensemble mean of the 26 other models. $\boldsymbol{\Sigma}_{\mathrm{y}}$ is then derived from the MAR fitted on the obtained residuals.

- the KCC method is applied using the inputs $\boldsymbol{y}$, $\boldsymbol{\Sigma}_{\mathrm{y}}$, $\boldsymbol{\mu}_{\mathrm{x}}$, $\boldsymbol{\Sigma}_{\mathrm{x}}$ to calculate projected changes constrained by the pseudo-observations, extracted from the SSP2-4.5 simulation.

- these 4 steps are repeated for each available member of the considered model, and for all available models.

I use the confusion matrix to estimate the reliability of the method for the short (2021-2040), mid (2041-2060) and long term (2081-2100) periods (Table 1). The constrained distributions contain the true values from the pseudo-observations in 80% up to 84% of the cases. The rate of false prediction (i.e. when the constrained distribution does not contain the true value while the unconstrained distribution do) remains very low, especially over the 2021-2040 period. Note that in about 10% of the cases, the true value is not contained in both the constrained and unconstrained distributions. This could be explained by a high/low sensitivity of some models to external forcings in the projections compared to what is predicted by the other (unconstrained) models.





**Table 1.** Confusion matrix relating the temperature projection constrained by the KCC method and the true value from pseudo-observations. Rates (in %) are computed from 249 members (from 27 models). Each rate is normalized by the number of ensemble members for each model to avoid giving too much weight to models with a large ensemble.

| Leadtime | 2021-2040 | | 2041-2060 | | 2081-2100 | |
|---|---|---|---|---|---|---|
| True value | in $p(\boldsymbol{x}\|\boldsymbol{y})$ | not in $p(\boldsymbol{x}\|\boldsymbol{y})$ | in $p(\boldsymbol{x}\|\boldsymbol{y})$ | not in $p(\boldsymbol{x}\|\boldsymbol{y})$ | in $p(\boldsymbol{x}\|\boldsymbol{y})$ | not in $p(\boldsymbol{x}\|\boldsymbol{y})$ |
| in $\Pi(\boldsymbol{x})$ | 78 | 2 | 81 | 6 | 76 | 11 |
| not in $\Pi(\boldsymbol{x})$ | 4 | 16 | 3 | 10 | 4 | 10 |

As a second performance criterion to assess the error on the amplitude of the constrained SSTs in the future over the WH, I use the continuous ranked probability skill score (CRPSS), defined as the relative error between the constrained distribution
$p(\boldsymbol{x}|\boldsymbol{y})$ and a given reference (Hersbach, 2000). Here, I quantify the added value of the constraint by considering the unconstrained projections as a benchmark. Figure 4 shows positive CRPSS values, associated with a reduction in error of 35%, 30% and 20% on average for the constrained projections compared to the raw CMIP6 projections for the short, mid and long term period, respectively. The added value of the method is stronger for the short term than for the long term period, as observations are temporally closer for the former. Overall, these results demonstrate that the results from the method are not overconfident
and that the constrained uncertainty ranges are reliable.





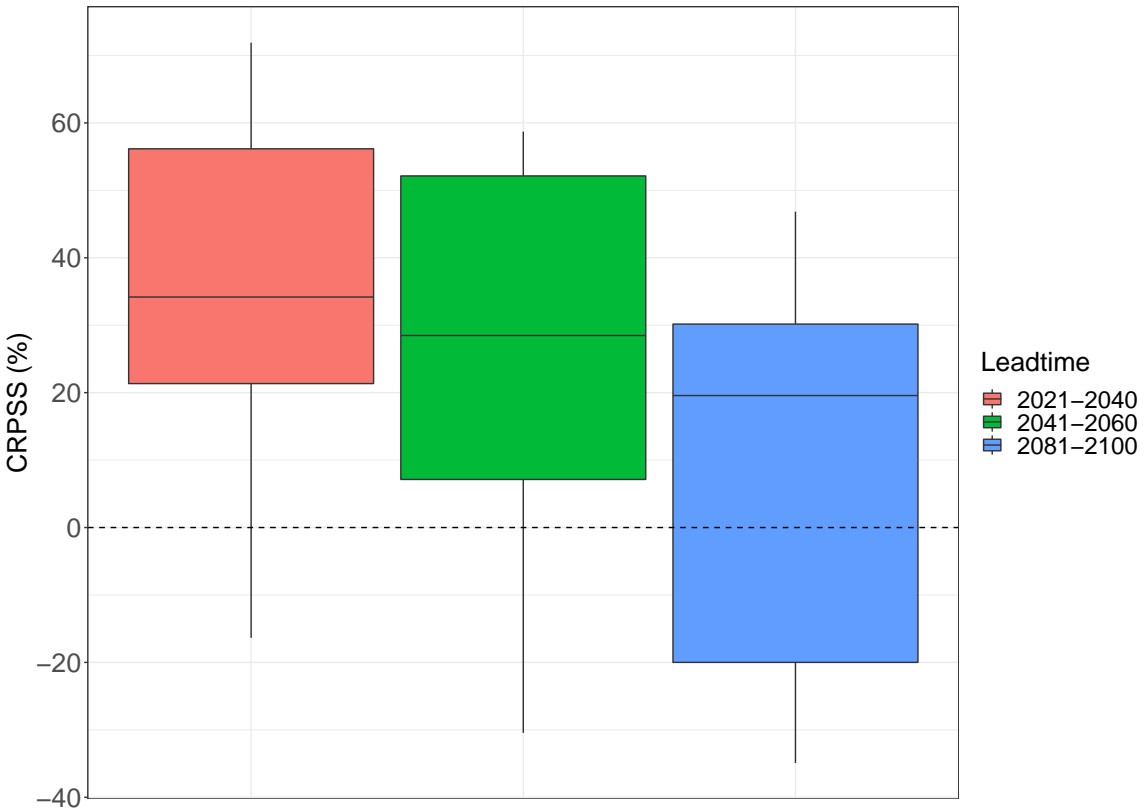

**Figure 4.** CRPSS for the constrained WH SST projections within the perfect model framework. The red, green and blue boxplots indicate the CRPSS distributions for different leadtimes. Calculation is made for all CMIP6 ensemble members (see Table A1). The top (bottom) of the box represents the 25th (75th) percentile of the distribution and the upper (lower) whisker represents the 95th (5th) percentile. Values are normalised by the number of members in each model. A CRPSS of 0 (dashed line) indicates the absence of added value of the method.

## 4 Conclusions

The temperature response to external forcings on the North Atlantic warming hole over the past and future period is estimated by the KCC statistical method based on kriging techniques, which combines climate models and observations. Consistent with the observations, an anthropogenic cooling is diagnosed by the method over the last decades (1951-2021) compared to the pre-industrial period. Although the anthropogenic response is clear, the respective contribution of the greenhouse gases and anthropogenic aerosols remains elusive. The quantification is in line with previous studies (Dagan et al., 2020; Chemke et al., 2020) that suggest that GHGs and aerosols have had compensating effects on the evolution of the WH. In the present study, the attribution of the temperature changes is based on only 12 models. Although the CMIP6 ensemble shows qualitatively consistent results, this large uncertainty points to the crucial need to increase the number of models running single forcing





experiments as done in DAMIP to better sample the model uncertainty. The method is also applied to CMIP6 projections, and is able to reduce the model uncertainty in the anthropogenic response by a factor of 3 over the historical period, and by a factor of 2 in the long-term projections. In particular, models projecting strong SST increase over the WH are excluded from the likely range constrained by the method.

This result has important implications for the estimation of the future changes in terms of teleconnection processes between
the North Atlantic and the continental climate, eg over Europe, North America, or the Sahelian monsoon. It would be interesting to re-evaluate the climate impacts of the North Atlantic SST variability in light of the constrained temperature ranges obtained in this study, for example in terms of the occurrence of extreme events or changes in atmospheric circulation.

A relevant perspective of these results is the potential constraint of the Atlantic Meridional Overturning Circulation (AMOC). Using the same approach as in this paper and to constrain directly future AMOC changes is challenging, due to the limited
number of observations monitored via the RAPID program (Frajka-Williams et al., 2021). Instead, taking advantage of the covariance between the AMOC and proxies based on SST and salinity over the North Atlantic (Zhang, 2017; Caesar et al., 2018) and applying the kriging method to these proxies is possible. This approach will allow to better estimate the uncertainty in the forced response included in these proxies, which could lead to revised estimates of the AMOC forced response over the past and future periods.

*Code and data availability.* The datasets generated and analysed during the current study are available at: https://doi.org/10.5281/zenodo.6952546. The HadCRUT5 dataset is available at https://www.metoffice.gov.uk/hadobs/hadcrut5/index.html. The HadSST4 dataset is available at https://www.metoffice.gov.uk/hadobs/hadsst4/index.html. The CMIP6 datasets are available at https://esgf-node.llnl.gov/projects/esgf-llnl/. All required programs to run the statistical method are in the associated KCC R package, which is available under a GNU General Public License, version 3 (GPLv3), at https://doi.org/10.5281/zenodo.5233947.





**Appendix A: List of the CMIP6 models**

**Table A1.** List of the available CMIP6 Models and the associated number of members in the simulations used to constrain the temperature time series.

| Model | Historical | Hist-nat | Hist-ghg | Hist-aer | SSP1-2.6 | SSP2-4.5 | SSP5-8.5 |
|---|---|---|---|---|---|---|---|
| ACCESS-CM2 | 3 | 3 | 3 | 3 | 3 | 3 | 3 |
| ACCESS-ESM1-5 | 20 | | 3 | 3 | 19 | 9 | 9 |
| BCC-CSM2-MR | 1 | 3 | 3 | 3 | 1 | 1 | 1 |
| CAMS-CSM1-0 | 2 | | | | 2 | 2 | 2 |
| CanESM5-CanOE | 3 | | | | 3 | 3 | 3 |
| CanESM5 | 50 | 50 | 50 | 50 | 50 | 50 | 50 |
| CESM2 | 6 | 3 | 3 | 2 | 5 | 6 | 5 |
| CESM2-WACCM | 3 | | | | 1 | 3 | 3 |
| CNRM-CM6-1 | 6 | 10 | 10 | 10 | 6 | 6 | 6 |
| CNRM-CM6-1-HR | 1 | | | | 1 | 1 | 1 |
| CNRM-ESM2-1 | 6 | | | | 5 | 6 | 5 |
| EC-Earth3 | 22 | | | | 7 | 22 | 8 |
| EC-Earth3-Veg | 5 | | | | 5 | 5 | 5 |
| FGOALS-f3-L | 3 | | | | 3 | 1 | 3 |
| FGOALS-g3 | 4 | 3 | 3 | 3 | 4 | 2 | 4 |
| FIO-ESM-2-0 | 3 | | | | 3 | 3 | 3 |
| HadGEM3-GC31-LL | 4 | 5 | 10 | 5 | 1 | 1 | 4 |
| INM-CM4-8 | 1 | | | | 1 | 1 | 1 |
| IPSL-CM6A-LR | 11 | 10 | 10 | 10 | 6 | 11 | 6 |
| MIROC6 | 50 | 50 | 3 | 10 | 50 | 3 | 50 |
| MIROC-ES2L | 10 | | | | 10 | 10 | 10 |
| MPI-ESM1-2-HR | 2 | | | | 2 | 2 | 2 |
| MPI-ESM1-2-LR | 10 | | | | 10 | 10 | 10 |
| MRI-ESM2-0 | 5 | 5 | 5 | 5 | 5 | 1 | 2 |
| NorESM2-LM | 3 | 3 | 3 | 3 | 1 | 3 | 1 |
| NorESM2-MM | 2 | | | | 1 | 2 | 1 |
| UKESM1-0-LL | 13 | | | | 13 | 5 | 5 |
| Total: 27 Models | 249 | 148 | 106 | 107 | 218 | 172 | 203 |





**Appendix B: Structure of the observation operator $H$**

The observation operator $H$ is a matrix of size $n_y \times n_x$. The constraint by both GMST and regional observations consists in setting the submatrices $H_{\mathrm{loc}} = I_{1870:2021}$ and $H_{\mathrm{glo}} = I_{1850:2021}$ and all other coefficients equal to zero (see block matrices in Eq. (B1)). This structure allows to extract the appropriate years from the vector $x$ (see Eq. (2)).

$$
\quad H = \left[ \begin{array}{c|c|c|c|c|c} H_{\mathrm{glo}} & 0 & 0 & 0 & 0 & 0 \\ \hline 0 & H_{\mathrm{loc}} & 0 & 0 & 0 & 0 \end{array} \right], \tag{B1}
$$

**Appendix C: Modelling of $\Sigma_{\mathrm{x}}$**

The DAMIP (or CMIP6) multimodel ensemble is used to derive a distribution of $x$, noted $\Pi(x) \sim \mathcal{N}(\mu_{\mathrm{x}}, \Sigma_{\mathrm{x}})$. $\mu_{\mathrm{x}}$ is the concatenated forced responses to all and specific external forcings mentionned in Eq. (2). From each model, I derive the forced responses following Ribes et al. (2021), i.e. 12 (27) vectors, from the DAMIP (CMIP6) ensemble. $\Sigma_{\mathrm{x}}$ is a variance-covariance

matrix of size $n_x \times n_x$ that describes the model spread. Using a sample covariance estimate has the side effect of producing a highly degenerated estimate for $\Sigma_{\mathrm{x}}$: while $\Sigma_{\mathrm{x}}$ is a $n_x \times n_x$ matrix, the rank of $\Sigma_{\mathrm{x}}$ is equal to 26 (since 27 CMIP6 models are being considered). While this choice could be debated, the KCC method can be run in this way, as $\Sigma_{\mathrm{x}}$ does not need being inverted to derive the parameters $\mu_p$ and $\Sigma_p$ of the posterior distribution.

In the Bayesian framework, $\Pi(x)$ is a first (probabilistic) estimate of $x$, which makes no use of observations, and is only

based on climate models. I update this estimate by incorporating the observational evidence provided by $y$. Following the Bayesian theory, the calculation of the posterior distribution $p(x|y)$ is required. A pre-requisite is to define the observational uncertainty, i.e., the covariance matrix associated with $y$.

**Appendix D: Modelling of $\Sigma_{\mathrm{y}}$**

$\Sigma_{\mathrm{y}}$ is estimated by using observed annual time series of GSAT and SST temperature over the historical period. First, I compute

the global observational residuals by subtracting the CMIP6 GSAT response (multimodel mean) to all external forcings from the observations $T_{\mathrm{glo}}^{\mathrm{obs}}$. Similarly, I derive regional residuals over the WH by subtracting the CMIP6 SST response from $T_{\mathrm{reg}}^{\mathrm{obs}}$. These residuals constitute a first estimate of global and regional internal variability, noted $\hat{\epsilon}_{y,glo,1}, \hat{\epsilon}_{y,reg,1}$, respectively.

I define $\Sigma_{\mathrm{y}}$ as a matrix of size $n_y \times n_y$ of the following form:

$$
\Sigma_{\mathrm{y}} = \left[ \begin{array}{c|c} \Sigma_{\mathrm{y,reg}} & \Sigma_{\mathrm{y,dep}} \\ \hline \Sigma_{\mathrm{y,dep}}' & \Sigma_{\mathrm{y,glo}} \end{array} \right], \tag{D1}
$$





where $\Sigma_{\mathrm{y,reg}}$ and $\Sigma_{\mathrm{y,glo}}$ are the covariance matrices modelling regional and global internal variability within $T_{\mathrm{reg}}^{\mathrm{obs}}$ and $T_{\mathrm{glo}}^{\mathrm{obs}}$, respectively. $\Sigma_{\mathrm{y,dep}}$ is the covariance matrix modelling the dependence between regional and global internal variability, i.e. $\epsilon_{\mathrm{y,reg}}$ and $\epsilon_{\mathrm{y,glo}}$.

To compute $\Sigma_{\mathrm{y}}$, I take into account decadal internal variability that may exist in the global (Parsons et al., 2020) and regional (Qasmi et al., 2017) observations, by using a mixture of two autoregressive processes or order 1 (AR1), hereafter MAR, as

done by Qasmi and Ribes (2021). The MAR formulation includes a fast (f) and a slow (s) components such that global internal variability $\epsilon_{y,glo}$ within the GMST residuals writes at a time $t$:

$$\begin{cases} \epsilon_{\mathrm{y,glo}}(t) & = \epsilon_{\mathrm{y,f,glo}}(t) + \epsilon_{\mathrm{y,s,glo}}(t), \\ \epsilon_{\mathrm{y,f,glo}}(t) & = \alpha_{\mathrm{f,glo}}\epsilon_{\mathrm{y,f,glo}}(t-1) + Z_{\mathrm{f,glo}}(t), \\ \epsilon_{\mathrm{y,s,glo}}(t) & = \alpha_{\mathrm{s,glo}}\epsilon_{\mathrm{y,s,glo}}(t-1) + Z_{\mathrm{s,glo}}(t), \end{cases} \tag{D2}$$

where the parameters $\alpha_{\mathrm{s,glo}}$ and $\alpha_{\mathrm{f,glo}}$ are the lag 1 coefficients of the AR1 processes, and $\alpha_{\mathrm{s,glo}} \geq \alpha_{\mathrm{f,glo}}$ by convention. $Z_{\mathrm{s,glo}}(t) \sim \mathcal{N}(0,\sigma_{\mathrm{s,glo}}^2)$ and $Z_{\mathrm{f,glo}}(t) \sim \mathcal{N}(0,\sigma_{\mathrm{f,glo}}^2)$ are white noises associated with the two AR1. The slow component is able

to generate a dependence on time scales of typically one decade, while the fast component accounts for interannual variability. The covariance matrix $\Sigma_{\mathrm{y,glo}}$ is filled according D2. The same assumptions are adopted to estimate the regional parameters and to compute $\Sigma_{\mathrm{y,reg}}$.

The initial estimate of $\Sigma_{\mathrm{y}}$, noted $\hat{\Sigma}_{\mathrm{y,1}}$ is solely based on the residuals $\hat{\epsilon}_{\mathrm{y,reg,1}}$ and $\hat{\epsilon}_{\mathrm{y,glo,1}}$ derived from the unconstrained forced response. This first estimate is likely flawed as the real (and unknown) forced response is not necessarily consistent with

the unconstrained forced response estimated by $\mu_{\mathrm{x}}$. In addition, as $\mu_{\mathrm{x}}$ can be by construction different from the best estimate of the constrained forced response $\hat{\mu}_{\boldsymbol{x,1}}$ (the mean of the posterior distribution $p(\boldsymbol{x}|\boldsymbol{y})$), the residuals $\hat{\epsilon}_{\mathrm{y,reg,1}},\hat{\epsilon}_{\mathrm{y,glo,1}}$ before constraint are not always coherent with the residuals $\hat{\epsilon}_{\mathrm{y,reg,2}},\hat{\epsilon}_{\mathrm{y,glo,2}}$ computed as the $\boldsymbol{y}-\hat{\mu}_{\mathbf{x,1}}$ difference.

Hence, in order to ensure an accurate estimation of internal variability in the constraint procedure, an iterative algorithm is applied to find the MAR parameters that fit the residuals from the constrained forced response:

$$\boldsymbol{y} - \boldsymbol{\mu} \xrightarrow{\text{residuals}} \hat{\boldsymbol{\epsilon}}_{\mathrm{y,1}} \xrightarrow{\text{constraint}} \hat{\boldsymbol{\mu}_1} \tag{D3}$$
$$\boldsymbol{y} - \hat{\boldsymbol{\mu}_1} \xrightarrow{\text{residuals}} \hat{\boldsymbol{\epsilon}}_{\mathrm{y,2}} \xrightarrow{\text{constraint}} \hat{\boldsymbol{\mu}_2}$$
$$\dots$$
$$\boldsymbol{y} - \hat{\boldsymbol{\mu}_{n-1}} \xrightarrow{\text{residuals}} \hat{\boldsymbol{\epsilon}}_{\mathrm{y,n}} \xrightarrow{\text{constraint}} \hat{\boldsymbol{\mu}_n}$$

where, for each iteration $n$, $\hat{\boldsymbol{\mu}_n}$ and $\hat{\boldsymbol{\epsilon}}_{\mathrm{y,n}}$ are estimates of the forced response and internal variability, respectively. The

termination criterion is based on the Frobenius norm $||.||_F$. Hence, I consider that the algorithm converges at the iteration $n$, i.e. that $\hat{\boldsymbol{\epsilon}}_{\mathrm{y,n}} \rightarrow \boldsymbol{\epsilon}_{\mathrm{y}}$, when the relative difference between $||\hat{\Sigma}_{\mathrm{y,n}}||_F$ and $||\hat{\Sigma}_{\mathrm{y,n-1}}||_F$ is inferior to 1 %, meaning also the MAR parameters values have converged. In practice, only 2 iterations are necessary in this study.



Initial condition large ensembles and long piControl simulations provide a nice sampling of internal variability, and could also be used to estimate this variability. However, I choose to not directly rely on it because of the huge discrepancies between models in terms of their simulated internal variability (Parsons et al., 2020). Qasmi and Ribes (2021) have illustrated this aspect with the piControl simulations from the CMIP6 models, including those used to build large ensembles. In all cases, the models do not converge to a consistent estimate of internal variability over the WH.

*Author contributions.* SQ developed the model code and wrote the manuscript.

*Competing interests.* The author declares that he has no conflict of interest.

*Acknowledgements.* This work was supported by the European Union's Horizon 2020 Research and Innovation Programme in the framework of the EUCP project (Grant Agreement 776613), the CONSTRAIN project (Grant Agreement 820829) and Météo-France. I thank Hervé Douville, Aurélien Ribes for fruitful discussions about this work. I thank the climate modeling groups involved in CMIP6 exercises for producing and making available their simulations. I thank the ETH Zurich for providing CMIP6 data through their cmip6-ng interface (http://dx.doi.org/10.5281/zenodo.3734128). The analyses and figures were produced with the R software (https://www.R-project.org/) and the NCAR Command Language Software (http://dx.doi.org/10.5065/D6WD3XH5).



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
