# Peer review of "Past and future response of the North Atlantic warming hole to anthropogenic forcing"

_Earth System Dynamics, 2022_

## Author Comment (AC1)

**Response to Referee #1**

I find this a very interesting and valuable statistical study comparing observed northern Atlantic sea surface temperature (SST) changes and model-simulated changes.

I am not a statistics expert and recommend that the statistical methodology should be reviewed by a suitable colleague, but I think I understand the basic approach. A key point of this is that no particular physical mechanism is studied (such as ocean circulation change), but rather simply the statistical relation of SST fields from various model runs (with different forcings) to the observed SST field. In this way we can find how various forcings relate to SST changes (though not by what mechanism) in the models, and whether that resembles what is observed in the northern Atlantic. I think it would be useful to make this a bit more clear right at the outset, because it confused me a bit while reading the introductory part of the paper, until I got to the Methods.

Thank you for these remarks. I have expanded the introduction to state more specifically that I was only interested in the statistical aspects and not the physical mechanisms for attributing the observed changes in SST over the historical period.

Important findings of this study in my view are:

- that aerosol forcing leads to the opposite SST response compared to observed. This should lay to rest the previous discussions of whether the Atlantic warming hole is caused by aerosol forcing.

- that the historical runs of CMIP6 with all forcings do not get the observed warming hole.

- that increasing greenhouse gases are the main reason for the observed warming hole.

If the author agrees with my assessment, I suggest bringing these conclusions out in somewhat clearer language in the paper, including the abstract.

I agree with the referee to put more emphasis on the results related to attribution. The proposed bullets are indeed important, yet, they are only referring to the unconstrained CMIP6 simulations. I have reworded them to emphasize the results from the statistical method, by mentioning that:

- The warming hole has an anthropogenic origin.
- The impact of the aerosols is an increase in SST which is opposed to the effect of GHGs which largely contribute to the cooling of the warming hole over the historical period.
- Large uncertainties remain in the quantification of the impact of each anthropogenic forcing.

I have added these elements in the manuscript, especially in the abstract, and I have developed them in the conclusion.

I recommend using fewer acronyms (like WH for warming hole) because it makes the paper hard to read.

I have reduced the number of acronyms as requested.

---

## Author Comment (AC2)

**Response to Referee #2**

Overall quality/General comments

The paper assess the SST response in the subpolar gyre region (the region is also known as warming hole) to different forcing and reduces the uncertainties in these responses using a kriging method. While I am not an expert in the statistics (maybe this should be looked at by another reviewer?), the method seems both appropriate and valid to me. It is not that easy to understand though, and I believe the paper would benefit from some clarifications (see specific comments). The method is furthermore used to reduce the uncertainty range of the future WH SST evolution and it is tested whether the results appear valid and indeed add an extra value to what was known before.

The paper does not look at the actual processes causing the different responses, but neither does it claim to do so, therefore, I think that is all right. It does however mean, that the informative value of the paper is not overly large, yet as the WH is a much discussed phenomena I still believe that the findings are valuable to the scientific community. Therefore, I would recommend publication once the comments listed below have been fully addressed.

Thank you for these remarks.

Specific comments

General: Regarding the method I am not sure why the author is assessing the responses to different forcing when assessing regional SST but not when looking at global GSAT? I see that he is mainly interested in the WH but then why consider GSAT at al (okay, I see the latter is explained in the text)l? Plus I wonder whether it can be a problem that air temperature and sea surface temperatures are put into one vector (as they have slightly different variability)?

Regarding the second question (I suppose that the first question is no longer relevant given the remark in brackets), it is not a problem to have these two variables in the same vector, because the covariance between the two is taken into account in the Sigma_x (for CMIP6 models) and Sigma_y (for observations) matrices used for the attribution and observational constraint calculations.

L 15 ff Drijfhout et al., 2012 did not compare the 1901-2021 period to the 1870-1900 period, please cite a different source

I have changed the reference.

LI 16-19 While the modifications in meridional heat transport are linked to the AMOC, the melting of ice has (so far) not been shown to have affected the AMOC, please rephrase

Done.

L 40 Kriging for Climate Change = KCC method? Please introduce abbreviation that is used later

I have added the acronym.

L 55 ff How many ensemble members are in the DAMIP ensemble and what models have been used? Please provide more information and state why you think that there are enough models/members in there to provide an adequate estimate of the response to the individual forcing (as I though the DAMIP ensemble is rather small…)

The list of models is provided in Table A1. Regarding the sensitivity of the estimate of the response to different forcings, the DAMIP ensemble may indeed seem small (106 members for 12 models), so I also used the available CMIP6 models (27 models, 249 members) to make the estimate of the GHG response using the 1%CO2 simulations to reconstruct a hist-GHG equivalent for each CMIP6 model (following Ribes et al 2020). The estimates made with this larger ensemble are consistent with those calculated from the DAMIP ensemble only (Figure 2 vs. Figure S9). I have added this clarification in the manuscript.

LI 69 I am not sure what you mean when you say The sample of the forced responses… do you mean a sample or do you mean the mean/average? Please elaborate

Here, the sample refers to the probability distribution built from the CMIP6 models.

LI 76 ff Can you explain why you expect that some parts of x are not observed in y? I would assume the full forced time series should be a part of y and if you can not observe it in y then because it is "cancelled out" by epsilon. But it seems I don't fully understand the method.

x contains the forced response from 1850 to 2100 (nx = 251) and y contains observations from 1850 to 2021 (ny = 172). epsilon is an estimate of internal variability over 1850-2021. The operator H is an observation operator of size ny × nx, which extracts the part of x that matches y, i.e., the forced response from 1850 to 2021. This is detailed in Appendix B.

L 79 Why is this remarkable? Please elaborate or leave this sentence out.

I have deleted this sentence.

LI 101 ff Is it correct that you assume that the CMIP6 mean response = the forced response (aka you cannot calculate the mean of the forced responses as you identify the forced response as the mean)? If yes, then how do you determine the covariance of the forced response? If not, please explain how you mean this.

For each CMIP6 model considered, I estimate the forced response in each of the "T" vectors shown in Eq (2). So, I estimate the forced response in GSAT, in mean SST over the warming hole, and also the response to specific forcings (i.e., NAT-only or GHG-only). I use all available members to make this calculation. As a result, I have a sample of 27 estimates of x -- 1 for each CMIP6 model considered.

Then, I compute the sample mean and variance over this sample of 27 vectors. These are our estimates for mu_x and Sigma_x.

This is detailed in Appendix C.

Ll 112-113 Why do you not use such a more complex hierarchical model? Do you believe that it is not necessary in this case?

I have removed this sentence as it was incorrect. The uncertainty is indeed included in the observations (200 members for HadCRUT5 and HadSST4). Sorry for this mistake.

L 121 Could you give 2-3 of those studies as a reference?

Done.

Fig. 1 Is there any reason why you only go until 2014 here, whereas later you consider the period until 2020? It would be nice if these were the same.

Unfortunately, DAMIP simulations only cover the 1850-2014 period (only few modeling centers have extended their simulations to 2020). These are shown in Fig.1. I have used a technique that uses the 1%CO2 simulations to extend several hist-GHG simulations to 2020, but it is only applicable for times series. Note that the NAT and OA estimates in Fig. 2 are not based on DAMIP simulations, so it is possible to consider the whole observed period until 2020.

Fig. 2 It would be good to mention in the caption that this refers to SST in the WH (not GSAT)

Done.

Ll 140 ff I am a little confused how the uncertainties in the responses to the ANT forcing can be so much smaller than those to the GHG and OA forcing even though the former consists of the latter (Fig.2)

This is due to the fact the estimate of the OA response is the result from the ALL - GHG difference. The ALL and GHG reponses are derived from the historical and hist-GHG simulations, respectively. Therefore, if there is a large uncertainty for the GHG estimate, there is also one for the OA response by construction. I have mentioned this in the description of the results.

L 174 I would remove slightly since whether >1 degree can be called slightly is debatable

Done.

L 191 Sorry, but what is the confusion matrix and why can you used it here?

The confusion matrix aims to quantify the coverage probabilities, derived from the number of cases for which the true value (from the pseudo-obs) :
- is included in both the constrained range p(x|y) and the unconstrained range Π(x) (true positive rate),
- not included in both the constrained and unconstrained ranges (true negative rate),
- is included in the constrained range, but not in the unconstrained range (false positive rate)

- is included in the unconstrained range, but not in the constrained range (false negative rate).

I have added this explanation in the manuscript.

LI 1922 But shouldn't this be the case in 90% of the cases, since you are considering the 5-95% interval?

If the method was perfect, I should indeed have a rate of 90%. The reasons I give in the manuscript (the differences in sensitivity between the pseudo-obs and the rest of the models) could explain the difference between the rates I get (arround 84%) and the target of 90%.

Technical corrections

Title forcing instead of forcings

L 143 Fig. 1d (not de)

L 166 verb missing in sentence

Done.

---

## Author Response (AR2)

**Response to Referee #2**

I read the manuscript only regarding the validity of the used statistical methods, but not regarding its application domain content. I found the overall method and details convincingly motivated and could not detect any mistakes.

Thank you for this comment and for reviewing the method.

I have only one request for improvement:

Line 212, "which is close to...": In view of the corresponding remarks by anoner reviewer, please replace this statement by a short discussion of the results of a binomial test where you *test* the hypothesis that the underlying probability of containment equals the expected 90%. I expect that the relative frequency of only 85% observed in your leave-on-out exercise will constitute a *significant* deviation from the expected 90%, at least if you assume all 200 (is this number correct?) trials to be independent. If you choose a smaller number of degrees of freedom because you consider the trials to be partially dependent, please justify your choice of degrees of freedom in that binomial test.

As suggested, I have conducted a binomial test with a chosen degree of freedom of 27. Although the number of pseudo-observations is 172 (see table A1), we cannot consider the members of the same model as completely independent from each other as they share the same forced response. Therefore, I chose to apply the assumption of independence to the models rather than the members. This choice is also debatable since many models share common components; nevertheless, it is much less harmful than considering the members of the same model as independent. In practice, for each model, the number of times the pseudo-observation is contained in the constrained range is weighted by the number of members of that model (e.g., by 1/50 for CanESM5, see caption of Table 1). In this framework, the binomial test indicates that a success rate of 80% remains compatible with an expected rate of 90% (p-value = 0.18). I have added this point to the manuscript.

Minor questions and corrections:

L.21: "has" --> "have"
L.41: "in this paper" appears twice

Fixed

L.45: Please explain shortly the motivation for calling this "Kriging" and the relationship to ordinary or Bayesian Kriging since that is nonobvious.

This term is used to refer to the interpolation aspect between observations and models. I added this clarification to the manuscript.

L.54, "median": I welcome taking a robust statistics approach here. But when taking the median rather than the mean, should one then not also take the root-mean-square (or even

mean absolute) difference from the median (!) as the corresponding estimate of variability? and how could one then treat covariance in such a robust statistics approach?

I do not fully understand this question. $\Sigma_y = \Sigma_{meas} + \Sigma_{iv}$ is the observation error covariance matrix, where $\Sigma_{meas}$ and $\Sigma_{iv}$ describe the measurement error and internal variability, respectively.
$\Sigma_{meas}$ is estimated as the sample covariance matrix over the 200-member ensemble of the HadSST4 dataset.
$\Sigma_{iv}$ is estimated using observed annual SST time series over the 1850-2021 period. I do not use the observed median to estimate the forced component, but the CMIP6 multi-model mean which provides a priori a better estimate because as you seem to say (at least as I understood it), the trend in the observations is polluted by internal variability. Hence it is difficult to separate the two components. Both errors due to the internal variability and the measurement uncertainty are taken into account in the calculation of $\Sigma_y$.

L.56, "the HadCRUT5 ensemble of 200 members": Are these 200 members in a one-to-one matching correspondence to the 200 HadSST4 members, so that covariance between SST and GSAT can be estimated?

The HadCRUT5 and HadSST4 ensembles have been generated independently, hence there is no relationship between the members.

L.98: "a some" --> "some"
L.112: "are to" --> "are used to"

Fixed.

L.150: "half of the observed cooling": Why is this so low, is the internal variability that large? Maybe point out that at least the confidence intervals of "Obs" and "constrained ALL" overlap.

L.183: "warmig" --> "warming"
L.193: "confidence these" --> "confidence in these"

Fixed.

L.200: "Sigma_y..." --> Is this done in the same iterative way (D3) as for the actual results?

Yes, I have added this precision.